# Flexible RSV Prefusogenic Fusion Glycoprotein Exposes Multiple Neutralizing Epitopes that May Collectively Contribute to Protective Immunity

**DOI:** 10.3390/vaccines8040607

**Published:** 2020-10-14

**Authors:** Nita Patel, Jing-Hui Tian, Rhonda Flores, Kelsey Jacobson, Michelle Walker, Alyse Portnoff, Mimi Gueber-Xabier, Michael J. Massare, Greg Glenn, Larry Ellingsworth, Gale Smith

**Affiliations:** Novavax, Inc. 21 Firstfield Road, Gaithersburg, MD 20878, USA; npatel@novavax.com (N.P.); jhtian@Novavax.com (J.-H.T.); rflores@Novavax.com (R.F.); Kjacobson@Novavax.com (K.J.); mwalker@Novavax.com (M.W.); aportnoff@Novavax.com (A.P.); mguebre-xabier@Novavax.com (M.G.-X.); MMassare@Novavax.com (M.J.M.); gglenn@Novavax.com (G.G.); lellingsworth@novavax.com (L.E.)

**Keywords:** respiratory syncytial virus, fusion glycoprotein, prefusogenic RSV F, cotton rat

## Abstract

Human respiratory syncytial virus (RSV) is a cause of lower respiratory tract infection in infants, young children, and older adults. There is no licensed vaccine and prophylactic treatment options are limited. The RSV fusion (F) glycoprotein is a target of host immunity and thus a focus for vaccine development. F-trimers are metastable and undergo significant rearrangements from the prefusion to a stable postfusion structure with neutralizing epitopes on intermediate structures. We hypothesize that vaccine strategies that recapitulate the breathable F quaternary structure, and provide accessibility of B-cells to epitopes on intermediate conformations, may collectively contribute to protective immunity, while rigid prefusion F structures restrict access to key protective epitopes. To test this hypothesis, we used the near full-length prefusogenic F as a backbone to construct three prefusion F variants with substitutions in the hydrophobic head cavity: (1) disulfide bond mutant (DS), (2) space filling hydrophobic amino acid substitutions (Cav1), and (3) DS, Cav1 double mutant (DS-Cav1). In this study, we compared the immunogenicity of prefusogenic F to prefusion F variants in two animal models. Native prefusogenic F was significantly more immunogenic, producing high titer antibodies to prefusogenic, prefusion, and postfusion F structures, while animals immunized with DS or DS-Cav1 produced antibodies to prefusion F. Importantly, prefusogenic F elicited antibodies that target neutralizing epitopes including prefusion-specific site zero (Ø) and V and conformation-independent neutralizing sites II and IV. Immunization with DS or DS-Cav1 elicited antibodies primarily to prefusion-specific sites Ø and V with little or no antibodies to other key neutralizing sites. Animals immunized with prefusogenic F also had significantly higher levels of antibodies that cross-neutralized RSV A and B subtypes, while immunization with DS or DS-Cav1 produced antibodies primarily to the A subtype. We conclude that breathable trimeric vaccines that closely mimic the native F-structure, and incorporate strategies for B-cell accessibility to protective epitopes, are important considerations for vaccine design. F structures locked in a single conformation restrict access to neutralizing epitopes that may collectively contribute to destabilizing F-trimers important for broad protection. These results also have implications for vaccine strategies targeting other type 1 integral membrane proteins.

## 1. Introduction

Human respiratory syncytial virus (RSV) is a cause of lower respiratory infection (LRTI) in young children and older adult populations. The disease burden is particularly high in developing countries with over 3 million hospitalizations and 50,000–70,000 deaths in young children in 2015 [1]. Children under 5 years of age are the most susceptible and account for 45% of RSV-related deaths, with the vast majority (>90%) in developing countries [1]. The RSV disease burden is also substantial in older adults with over 1 million infections, 300,000 hospitalizations, and over 10,000 in-hospital deaths worldwide in 2015 [2]. Although RSV is the cause of significant worldwide disease burden, there are no licensed vaccines and palivizumab (Synagis^®^) is the only licensed prophylaxis for prevention of RSV in high-risk newborns. Unlike influenza where antibodies to hemaggulinin is a correlate for protective immunity, there is no established correlate of protective for RSV [3,4,5].

RSV is a negative-strained, enveloped RNA virus in the *Pneumoviridae* family. The fusion (F) glycoprotein is a major component of the virus envelope. RSV F protein is conserved between human RSV A and B subtypes (>90% amino acid identity) with shared neutralizing epitopes, is essential for infection, and is a major target of host immune defense [6,7,8,9,10]. RSV F is a type 1 integral membrane protein produced as a 70 kDa inactive precursor (F0). Unique to RSV, the F glycoprotein has two cleavage sites at positions R109 (site I) and R136 (site II) that are processed by host cell furin-like proteases. Removal of the intervening 27 amino acid peptide (p27) generates a small F2 subunit and a larger F1 subunit [11,12,13,14]. The F1 subunit contains the fusion peptide (FP) on the N-terminus, fusion machinery heptad repeats A and B (HRA and HRB), and the transmembrane (TM) domain and cytoplasmic tail (CT) on the C-terminus. The F2 subunit contributes to fusogenicity, and contains a single heptad repeat C (HRC) [7,15,16,17,18]. F2/F1 are covalently linked by two disulfide bonds to form a protomer. Three F2/F1 protomers associate through weak non-covalent interprotomeric bonds to form the functional F-trimer. Trimers are highly flexible and transiently open and close, dissociate, and monomerize within the lipid bilayer, exposing a range of conformations recognized by B-cell receptors [19,20].

Prefusion F undergoes significant rearrangements to a stable postfusion F during attachment and fusion. Events triggering F-protein rearrangement are not understood, although elevated temperatures promote spontaneous rearrangement of the heptad repeats (HRA, HRB and HRC) to form the fusogenic six-helix bundle (6HB) that releases the FP from the hydrophobic cavity. FP insertion into the host membrane is essential for alignment of the virus and host cell membranes, fusion, pore formation, and release of the virus genome into the host cell [21]. Amino acid substitutions in the hydrophobic cavity of RSV F stabilize the protein in the prefusion conformation. Disulfide-bond mutant DS (S155C and S290C), Cav1 with cavity-filling hydrophobic amino acids at positions S150F and V207L, and double mutant DS-Cav1 have been extensively characterized and shown to bind neutralizing monoclonal antibodies (mAbs) [22,23]. Prefusion F has been proposed as a superior vaccine candidate since antibodies bind antigenic sites zero (Ø) and V with high affinity, although there is no clinical evidence supporting protective efficacy [24].

Monoclonal antibodies (mAbs) derived from convalescent patients have provided insight into the plasticity of the humoral immune response to subtle conformational changes in the F structure. For instance, human neutralizing mAbs D25, AM22, and RSD5 all bind prefusion-specific antigenic site Ø with high affinity [22,25]. Crystal structures of prefusion DS (Protein Data Base (PDB) 4MMQ) and DS-Cav1 (PDB: 4MMS), in complex with D25, AM22, and RSD5 antigen binding fragments (Fabs), show these Fabs bind site Ø at three distinct angles [26,27]. Gillman et al. [19] have recently compared binding of human mAbs CR9501 and hRSV90 to antigenic site V. Crystal structures of CR9501 Fab in complex with antigenic site V shows the Fab is rotated by 60° compared to hRSV90 and CDR makes contact with F2. Significantly, CR9501 binding destabilizes F-trimers and promotes disassembly [19]. Similarly, mAbs palivizumab and motavizumab bind sites II at distinct angles and with different affinity [9,28,29]. We recently reported that murine mAb R4.C6 binds a quaternary epitope consisting of antigenic site II on one protomer and site IV on a neighboring protomer [28]. Collectively, these observations demonstrate that B-cells recognize a wide range of intermediate trimeric and monomeric F structures. These observations and others suggest trimeric vaccine candidates that mimic the breathable native F structure, allowing B-cell accessibility to sequestered epitopes, not only blocking receptor binding but, more importantly, may destabilize the trimeric structure.

We have described the development of a RSV prefusogenic F constructed from the near full-length RSV/A2 F protein in which furin cleavage site II was mutated to retain the native p27 with a truncated FP (ΔFP) on the N-terminus of F1. F1/F2 protomers assemble as trimers and 3–6 F-trimers spontaneously associate to form 35–40 nm nanoparticles. Prefusogenic F nanoparticles are immunogenic and protective in cotton rats [30,31], elicit maternal antibodies that protect newborn infant baboons against pulmonary RSV challenge [32], and are safe and immunogenic in women of childbearing age (NCT02624947) [33] and older adults [34]. Here, we used prefusogenic F as a backbone to assess the effects of introducing prefusion stabilizing mutations (DS, Cav1, DS-Cav1) upon the immunogenicity of prefusogenic F. Overall, we find that prefusogenic F was significantly more immunogenic than rigid DS or DS-Cav1 variants. Importantly, immunization with prefusogenic F elicited a broad range of antibodies that bound prefusion-specific antigenic sites Ø and V as well as serotype-conserved neutralizing sites II and IV. Prefusion DS and DS-Cav1 variants were less immunogenic and elicited antibodies primarily to sites Ø and V with little or no antibodies to other neutralizing F epitopes. These findings suggest that immunization with breathable F candidate vaccines results in a broad range of antibodies to neutralizing sites that may collectively contribute to protection against RSV. These results also have implications for vaccine strategies targeting other type 1 integral membrane proteins.

## 2. Materials and Methods

### 2.1. Cell Lines, Viruses, Synthetic Peptides, and Monoclonal Antibodies (mAbs)

HEp-2 (ATCC, CCL-23, Manassas, VA, USA) cells were maintained in minimal essential medium (MEM) with Earle’s salts and L-glutamine (Gibco Laboratories, Gaithersburg, MD, USA), 10% fetal bovine serum (FBS; HyClone, Logan, UT, USA), and antibiotics (Life Technologies, Grand Island, NY, USA). RSV/A Long challenge stock (lot KJ060319, 2.21 × 10^7^ plaque forming units (pfu) mL^−1^) was propagated in HEp-2 cells and produced by Sigmovir Inc. (Gaithersburg, MD, USA). Synthetic p27 peptide (RARRELPRFMNYTLNNPKKTNVTLSKKPKRRF) was produced by Thermo Fisher, Inc. Palivizumab (site II) was obtained from MedImmune, Inc. (Rockville, MD, USA). hRSV90 (site VIII, also referred to as site V) [35] was provided Dr. James Crowe (Vanderbilt Vaccine Center, Vanderbilt University, Nashville, TN). D25 (site Ø) [8,22,23] was purchased from Creative BioLabs (Shirley, NY, USA). Mouse mAbs R1.42 (site IV) and R7.10 (p27) were produced by Novavax, Inc. [28,31]. World Health Organization (WHO) human RSV Reference Standard was purchased from the National Institute of Biological Standards and Control ((NIBSC) code 16/284, 2000 IU mL^−1^, Potters Bar, Hertfordshire, UK).

Other reagents. RSV F DS-Cav1-293 reagent was produced by transient transfection of Expi293F cells (Invitrogen, Carlsbad, CA, USA) with pcDNA 3580 (Invitrogen) encoding the ectodomain of the RSV F gene mammalian condon optimized with the DS (S55C and S290C) and the Cav1 (S190F and V207L) mutations and a C-terminal T4 fibritin trimerization motif and 6xHisTag. Expi293F cells were transfected with Exipifectamine 293 transfection reagent following the manufacturer’s procedure (Invitrogen, Carlsbad, CA, USA). Prefusion F (BV2129) reagent was constructed from the prefusogenic F (BV1184) backbone with stabilizing triple mutations at positions N67I, S215P and E487Q [17] and produced in *Spodoptera frugiperda* (Sf9) insect cells.

### 2.2. RSV F Glycoprotein Constructs

Synthetic RSV F genes were cloned into the pFasBac1 (Invitrogen) downstream of the *AcMNPV* polyhedron promoter (GeneArt, Regensburd, DE, USA). RSV F transgenes were cloned into recombinant baculovirus (BV) using the Bac-to-Bac BV system [30]. Sf9 cells (Invitrogen, Grand Island, NY, USA) were cultured in serum-free medium. RSV F proteins were constructed from the near full-length RSV/A2 F gene sequence (Genbank Accession No. U63644) encoding 574 amino acids. The constructs were codon optimized for expression in insect cells. To produce the prefusogenic F construct (BV1184), furin cleavage site II was mutated (KKRKRR → KKQKQQ) to be protease resistant and the adjacent 10 amino acids deleted (F137-V146) from the fusion peptide (ΔFP) on the N-terminus of the F1 subunit and the native transmembrane (TM) and cytoplasmic tail (CT) retained on the C-terminus of the F1 subunit (GenBank Accession No. MN125707) [29,31,36]. Three prefusion F constructs were generated using the prefusogenic F (BV1184) backbone: (1) RSV F DS (BV2267) was produced by introducing disulfide bond (DS) at positions S155C and S290C within the F1 subunit; (2) RSV F Cav1 (BV2279) construct was generated by introducing hydrophobic amino acid substitutions at positions S190F and V207L; and (3) RSV F DS-Cav1 (BV2280) was constructed with the DS and Cav1 substitutions [22]. Postfusion F (BV2128) was constructed with the native furin cleavage sites I and II and the truncated fusion peptide (ΔFP). N-terminal TM and CT domains were deleted and replaced with a 6-histidine tag (6-H) [30,31].

### 2.3. Purification of RSV F Proteins

Prefusogenic F (BV1184), postfusion F (BV2129), prefusion F DS (BV2267), Cav1 (BV2279), and DS-Cav1 (BV2280) were produced by infecting Sf9 cells (MOI = 0.5 pfu/cell) with recombinant BV for 67 h at 27 °C. Cells were harvested by centrifugation and detergent-lysed with 25 mM Tris-HCL (pH 8.0), 50 mM NaCl, 0.5% (*v*/*v*) Tergitol™ NP-9, and 2.0 µg mL^−1^ leupeptin (Sigma-Aldrich, St. Louis, MO, USA), then RSV F proteins were extracted from cell membranes. RSV F proteins were purified by a combination of anion exchange, lentil lectin affinity, and cation ion exchange chromatography [30]. For soluble postfusion F (BV2128), Sf9 cells were infected with recombinant BV for 67 h and supernatants were clarified by centrifugation. RSV F (BV2128) was purified with an immobilized metal affinity column (IMAC) and ion exchange chromatography [31]. Purified RSV F proteins (0.33-1.5 mg mL^−1^) were formulated in 25 mM phosphate buffer (pH 6.8), 150 mM NaCl, 0.032% (*v*/*v*) polysorbate-80 (PS80), and 1% (*w*/*v*) histidine.

### 2.4. Antigenic Site-Specific mAb Binding to RSV F Proteins Determined by ELISA

Site-specific mAbs binding to recombinant RSV F proteins was determined by ELISA. Purified prefusogenic F (BV1184) and prefusion F proteins DS (BV2267), Cav1 (BV2279), DS-Cav1 (BV2280), and DS-Cav1-293 were coated (2.0 µg mL^−1^) in 96-well microtiter plates overnight at 4 °C. Following coating, the plates were blocked with 2% milk blocking buffer (Quality Biologicals, Gaithersburg, MD, USA). Primary mAbs D25 (site Ø), hRSV90 (site V), palivizumab (site II), R1.42 (site IV), and R7.10 (p27) were serially diluted (1.0 µg mL^−1^ to 0.01 ng mL^−1^). WHO human RSV reference antiserum (RSV Ref Std) was serially diluted 10^−2^ to 10^−8^. Plates were incubated for 2 h at room temperature and washed with phosphate buffered saline containing 0.05% Tween (PBS-T). Horseradish peroxidase (HRP)-conjugated goat anti-human IgG (2040-05, Southern Biotech) or HRP-conjugated goat anti-mouse IgG (1030-05, Southern Biotech) secondary antibodies were added to the wells. After 1 h, the wells were washed with PBS-T and 3,3′,5,5′-tetramethylbenzidine peroxidase substrate (TMB, T0440-IL, Sigma, St Louis, MO, USA) added to the wells. Reactions were stopped with TMB stop solution (TSB999, Scytek Lab, Logan, UT, USA). Plates were read at an optical density (OD) of 450 nm with a SpectraMax plus plate reader (Molecular Devices, Sunnyvale, CA, USA). EC_50_ values were calculated by 4-parameter fitting using Prism 7.05 software.

### 2.5. Design of Animal Studies

*Animal ethics statement.* Noble Life Sciences (Sykeville, MD, USA) performed the murine immunogenicity study. Noble Life Sciences is accredited by the Association for Assessment and Accreditation of Laboratory Animal Care (AAALAC International). Sigmovir Biosystems, Inc. (Rockville, MD, USA) performed the cotton rat immunogenicity and RSV challenge study. Sigmovir is licensed by the United States Department of Agriculture (USDA permit 51-R-0091) and the Office of Laboratory Animal Welfare (OLAW permit A4642-01). The cotton rat study was conducted in accordance with appropriate sections of the National Research Council Guide for the Care and Use of Laboratory Animals and the Animal Welfare Act Regulations. Both studies were performed in accordance with Institutional Animal Care and Use Committee (IACUC) approved protocols.

*Murine study design.* Female BALB/c mice (7–9 weeks old, 17–22 g, N = 10 per group) were immunized by intramuscular (IM) injection with 1.0 µg of RSV F prefusogenic F, prefusion F variants (DS, Cav1, DS-Cav1) or postfusion F adjuvanted with 30 µg aluminum phosphate (Adju-Phos^®^, Brenntag Biosector, Frederikssund DK) administered in 2 doses spaced 21 days apart (day 0, 21). A separate group (N = 10) received a single intranasal (IN) challenge with 20 µL of 10^6^ pfu RSV/A Long (10 µL each nare) on study day 0. Serum was collected for analysis from individual animals on study day 35 (14 days after the second immunization).

*Cotton rat study design.* Female cotton rats (4–6 weeks old, 50–65 g) were randomly assigned to groups (N = 6 per groups) and immunized by IM injection with 1.0 µg of the indicated RSV F protein with 30 µg aluminum phosphate (lot 9496, Brenntag Biosector) administered in 2 doses spaced 21 days apart (day 0, 21). A control group (N = 6) received placebo (non-immunized) and was used as a positive RSV challenge control. A separate group (N = 6) received a single IN challenge with 100 µL of 10^6^ pfu RSV/A Long divided between nares on study day 0. Serum was collected via the orbital plexus 21-days after the second immunization (day 42) and stored at −20 °C until analyzed. Animals were challenged with 100 µL of 10^5^ pfu RSV/A Long (50 µL each nare) 21 days after the second immunization (day 42). Nose and lung homogenates were prepared 4 days post-challenge (day 46) and virus load determined by a plaque assay.

### 2.6. Anti-RSV F Glycoproteins IgG ELISA

An ELISA was used to determine anti-RSV F IgG titers. Briefly, 96-well microtiter plates (Thermo Fisher Scientific, Rochester, NY, USA) were coated with 2.0 µg mL^−1^ of prefusogenic F (BV1184), prefusion F (BV2129), DS-Cav1 (BV2280), or postfusion F (BV2128). Plates were washed with PBS-T and blocked with blocking buffer (Quality Biologicals, Gaithersburg, MD, USA). Serum samples were serially diluted (10^−2^ to 10^−8^) and the plates incubated at room temperature for 2 h. Following incubation, plates were washed with PBS-T and HRP-conjugated goat anti-mouse IgG or anti-rat IgG (Southern Biotech, Birmingham, AL, USA) added for 1 h. Plates were washed with PBS-T and TMB peroxidase substrate (T0440-IL, Sigma, St Louis, MO, USA) was added. Reactions were stopped with TMB stop solution (ScyTek Laboratories, Inc. Logan, UT). Plates were read at OD 450 nm with a SpectraMax plus plate reader (Molecular Devices, Sunnyvale, CA, USA) and data analyzed with SoftMax software. EC_50_ values were calculated by 4-parameter fitting using GraphPad Prism 7.05 software (San Diego, CA, USA). Individual animal anti-RSV F protein IgG titers and group geometric mean titers (GMT) and 95% confidence interval (±95% CI) were plotted.

### 2.7. RSV Neutralizing Assay

Mouse and cotton rat serum samples were serially diluted in 96-well microtiter plates. Palivizumab and human RSV reference standard (NIBSC 16/284, Potters Bar, Hertfordshire, UK) were used to generate standard curves. RSV/A Long or RSV/B 18537 fifty-percent tissue culture infective dose (TCID_50_) of 200–350 was added to the wells and incubated for 2 h at 37 °C in 5% CO_2_. Following adsorption, wells were seeded with 2.5 × 10^4^ HEp-2 cells and cultured for 4 days at 37 °C in 5% CO_2_. After incubation, the cultures were washed and air dried. To detect RSV infection, the cultures were treated with mouse mAb (RSV5H5) anti-RSV M2-1 (ab94805, Abcam, Cambridge, MA, USA). HRP-conjugated goat anti-mouse IgG second antibody (catalog number 1030-05, Southern Biotech, Birmingham, AL, USA) was added. TMB substrate was added to the wells and the plates read at OD 450 nm with a plate reader. Neutralizing titers were calculated relative to the human RSV reference standard (2000 IU mL^−1^) or palivizumab standard (4600 IU mL^−1^) using a 4-parameter fit using GraphPad Prism 7.05 software. Individual animal titers were plotted with the group GMT ± 95% CI indicated.

### 2.8. Prefusion and Postfusion F Specificity of Antibodies Determined by RSV F Competitive ELISA

Levels of serum antibodies specific for prefusion and postfusion F structures were determined with a competitive ELISA. Briefly, 96-well plates were coated with 2.0 µg mL^−1^ prefusion F (BV2129) overnight at 4 °C. Immunized cotton rat serum or RSV reference antiserum was 3-fold serially diluted and mixed (1:1 ratio) with 20 µg mL^−1^ of prefusion F (BV2129) or postfusion F (BV2128) for 1 h at room temperature. Following incubation, 100 µL of serum/protein mixture was transferred to the prefusion F-coated microtiter wells. The mixture was incubated for 2 h at room temperature and the wells and washed with PBS-T. HRP conjugated goat anti-rat IgG or anti-human IgG was added followed by addition of TMB substrate. Plates were read at OD 450 nm with a plate reader and the EC_50_ values calculated by 4-parameter curve fitting with GraphPad Prism software (San Diego, CA, USA).

### 2.9. Prefusion and Postfusion F Specificity of Neutralizing Antibodies Determined by Competitive ELISA

Levels of neutralizing antibodies bound by prefusion and postfusion F structures were determined with a competitive ELISA. Cotton rat serum was serially diluted in 96-well microtiter plates and 5 µg mL^−1^ prefusion F (BV2129) or postfusion F (BV2128) added for 1 h at 37 °C followed by addition of RSV/A (TCID_50_ = 200–300) for 2 h at 37 °C. Following incubation, 2.5 × 10^4^ HEp-2 cells were added to the wells and the plates incubated at 37 °C and 5% CO_2_ for 4 days. After incubation, the cultures were washed and air dried. Virus replication was detected with the mouse mAb RSV5H5 anti-RSV M2-1 (ab94805, Abcam, Cambridge, MA, USA) followed by addition of HRP-conjugated goat anti-mouse IgG secondary antibody (catalog number 1030-05, Southern Biotech, Birmingham, AL, USA). TMB substrate was added and the plates read at OD 450 nm with a plate reader. Percent neutralization was calculated compared to the virus control cultures (no serum).

### 2.10. Antigenic Site-Specific Antibody Binding Determined by ELISA

A competitive antibody binding assay was used to determine the levels of antibodies in mouse and cotton rat serum that competed the binding of RSV F site-specific mAbs [30]. Briefly, monoclonal antibodies D25 (site Ø), palivizumab (site II), R1.42 (site IV), and R7.10 (p27) were biotinylated using the manufacturer’s protocol (Pierce, Rockford, IL, USA). For sites II, IV and p27 determinations, microtiter plates were coated with 2.0 µg mL^−1^ prefusogenic RSV F protein (BV1184). For site Ø determinations, plates were coated with 2.0 µg mL^−1^ prefusion RSV F protein (BV2129). After overnight incubation, the plates were washed with PBS-T and blocked with 2% milk blocking buffer for 1 h at room temperature. Immune serum was 2-fold serially diluted and spiked with 10 ng mL^−1^ biotinylated D25, palivizumab, R1.42, or 50 ng mL^−1^ R7.10. Biotinylated antibodies without immune serum were used as a no competition control. After 2 h at room temperature, the plates were washed with PBS-T and 100 µL diluted HRP conjugated Streptavidin (e-Bioscience, San Diego, CA, USA) was added for 1 h and plates washed with PBS-T. TMB substrate was added, color developed, and reaction stopped with TMB stopping buffer. Plates were read at OD 450 nm with a plate reader. Data was analyzed with SoftMax Pro software (Molecular Devices). Competing antibody concentration was determined from the RSV reference standard curve by interpolating the concentration of the reference standard that corresponds to 50% competition. Antigenic site-specific antibody levels (µg mL^−1^) were plotted for individual animals and the group GMT ± 95% CI indicated.

### 2.11. Plaque Assay

Nose and lung tissue homogenates were prepared from cotton rats 4 days following IN challenge with RSV/A Long. Homogenates were clarified by centrifugation. Confluent HEp-2 monolayers were infected in duplicate with 50 µL per well of serially diluted homogenate samples in 24-well plates. After 1 h at 37 °C in 5% CO_2_, 0.75% methylcellulose medium overlay was added and the cells were cultured for 6 days. Cells were fixed with 10% neutral formalin and stained with 0.01% crystal violet solution (1.5 mL/well). Plaques were counted and titers expressed as plaque forming units per gram (pfu g^−1^) of nasal or lung homogenate. Virus titers were plotted for individual animals and the group GMT ± 95% CI indicated.

### 2.12. Statistical Analysis

Statistical comparisons between paired groups was performed with a Student’s *t*-test and one-way ANOVA using GraphPad Prism 7.05 software (San Diego, CA, USA).

## 3. Results

### 3.1. RSV F Glycoproteins Proteins

We have described a prefusogenic RSV F (BV1184) vaccine candidate constructed from the near full-length RSV/A2 F-glycoprotein that form nanoparticles consisting of 3–5 F-trimers [30,31]. Using the prefusogenic F as a backbone, prefusion constructs were produced by introducing mutations within the hydrophobic cavity to stabilized the prefusion F structure as described by McLellan et al. [22,23]. RSV F DS (BV2267) was constructed with the disulfide bond mutation (DS) at positions S155C and S290C; RSV F Cav1 (BV2279) with amino acid substitutions at positions S190F and V207L; and RSV F DS-Cav1 (BV2280) double mutant (Figure 1A,B).

### 3.2. Antigenic Site-Specific Binding of mAbs to RSV F Proteins by ELISA

A panel of human and murine mAbs targeting RSV F neutralizing sites was used to characterize the antigenic profile of the RSV F constructs. Using an ELISA method, prefusion-stable RSV F DS (BV2267), Cav1 (BV2279), DS-Cav1 (BV2280), and DS-Cav1-293 were bound by prefusion-specific mAbs D25 (site Ø) and hRSV90 (site V) at low antibody concentrations (EC_50_ = 1.07–1.66 ng mL^−1^), indicating neutralizing sites Ø and V were intact on these prefusion constructs. Prefusogenic F nanoparticles (BV1184) were also bound by mAbs D25 and hRSV90, although at higher antibody concentrations (EC_50_ = 47.40 ng mL^−1^ and 26.80 ng mL^−1^, respectively) suggesting these prefusion sites are present on prefusogenic F, although at lower levels (Figure 2A,B). Humanized palivizumab [9] and murine R1.42 mAbs [28] were used to detect antigenic sites II and IV, respectively. Prefusogenic F and prefusion F variants (DS, Cav1, DS-Cav1 and DS-Cav1-293) bound palivizumab (ED50 = 1.0–3.0 ng mL^−1^) and R1.42 (EC_50_ = 2.4–6.1 ng mL^−1^) at low antibody concentrations indicating sites II and IV were present on prefusogenic F and the prefusion F variants (Figure 2C,D). Prefusogenic F and prefusion DS, Cav1, and DS-Cav1 variants also bound mAb R7.10 at low concentrations (EC_50_ = 2.7–3.17 µg mL^−1^) consistent with the p27 being preset on these F-trimers. DS-Cav1-293 bound R7.10 at a higher concentration (ED_50_ = 22.1 µg mL^−1^) compared to the other constructs, suggesting p27 may be less available to antibody binding (Figure 2E). All RSV F structures bound WHO RSV reference antibodies with comparable affinity (EC_50_ = 1.12E-04 to 3.97E-04) (Figure 2F). These results indicate that antigenic sites Ø, V, II, IV, and p27 were present on prefusogenic F and the DS, Cav1, DS-Cav1 and DS-Cav-293 variants.

### 3.3. Immunogenicity of RSV F Proteins in Mice

The immunogenicity of prefusogenic F was compared to prefusion DS, Cav1, and DS-Cav1 F variants in mice. Animals immunized with prefusogenic F (BV1184) had significantly higher (*p* < 0.0001) antibody titers to prefusogenic F BV1184 (GMT = 384,547) compared to titers in animals immunized with DS (GMT = 2736) or DS-Cav1 (GMT = 1038). Animals immunized with prefusogenic F also had significantly higher (*p* ≤ 0.0007) antibody titers to prefusion F BV2129 (GMT = 223,670) compared to animals immunized with prefusion F variants DS (GMT = 20,527) or DS-Cav1 (GMT = 19,195) (Figure 3A,B). Immunization with Cav1 (BV2279) also elicited high titer antibodies to prefusogenic F BV1184 (GMT = 161,953) and prefusion F BV2129 (GMT = 109,422) that were not significantly different from titers upon immunization with prefusogenic F (Figure 3A,B).

Levels of serum antibodies targeting neutralizing epitopes were determined by competitive ELISA using site-specific mAbs. Mice immunized with Cav1 had the highest levels of antibodies (GMT = 7.8 µg mL^−1^) competitive with D25 (site Ø), while serum from mice immunized with prefusogenic F, DS, or DS-Cav1 variants had significantly lower (*p* ≤ 0.034) levels of antibodies that competed D25 binding (GMT = 0.7–1.4 µg mL^−1^). These results suggest that antigenic site Ø was poorly immunogenic in mice regardless of the immunogen (Figure 3C).

Mice immunized with prefusogenic F and Cav1 had significantly higher (*p* < 0.0001) levels of antibodies that competed the binding of palivizumab to site II (GMT = 620.0 and 224.5 µg mL^−1^, respectively), while mice immunized with prefusion F DS or DS-Cav1 had little or no antibodies targeting site II (GMT = 4.0 and 2.5 µg mL^−1^, respectively) (Figure 3D). Similarly, animals immunized with prefusogenic F or Cav1 had significantly (*p* < 0.0001) elevated levels of antibodies that competed binding of R1.42 to site IV (GMT = 676.9 and 204.4 56.0 µg mL^−1^, respectively), while animals immunized with DS or DS-Cav1 had no detectable antibodies that competed binding of R1.42 to site IV (Figure 3E). Animals immunized with postfusion F also had low levels of antibodies competitive with palivizumab for site II (GMT = 33.8 µg mL^−1^) and variable levels of antibodies competitive with R1.42 for site IV (GMT = 46.7 µg mL^−1^) (Figure 3D,E).

Antibodies targeting p27 were also determined. Mice immunized with prefusogenic F or Cav1 had significantly higher (*p* < 0.0001) levels of antibodies that competed with R7.10 to the p27 peptide (GMT = 112.2 and 80.9 µg mL^−1^, respectively) (Figure 3F). In contrast, immunization with DS or DS-Cav1 variants elicited no measurable antibodies to p27 (GMT < 0.35 µg mL^−1^), suggesting that p27 was not accessible to B-cells in DS and DS-Cav1 stabilized F (Figure 3F). Animals immunized with the prefusogenic F, DS, or DS-Cav1 variants had similar levels of RSV/A Long neutralizing antibodies (GMT = 86.2 to 225.4 IU mL^−1^) (Figure 3G). Immunization with postfusion F (BV2128) produced significantly lower (*p* = 0.011) levels of neutralizing antibodies. Taken together, these results demonstrate that prefusogenic F was significantly more immunogenic than prefusion F DS or DS-Cav1. Antigenic site Ø was poorly immunogenic and elicited little or no antibodies competitive with D25 in mice regardless of the immunogen. Prefusogenic F and Cav1 were equally immunogenic and elicited antibodies targeting sites II and IV, while immunization with prefusion stable DS and DS-Cav1 had little or no antibodies to antigenic sites II or IV or p27, suggesting these sites were not accessible to humoral immune response on the prefusion-stable F structures.

### 3.4. Immunogenicity of RSV F Proteins in Cotton Rats

The immunogenicity of prefusogenic and prefusion F variants was also compared in cotton rats. Animals immunized with prefusogenic F had significantly higher (*p* < 0.0001) titers of antibodies to prefusogenic F BV1184 (GMT = 285,691) compared to titers in animals immunized with DS (GMT = 18,361) or DS-Cav1 (GMT = 4590) (Figure 4A). Animals immunized with prefusogenic F also had significantly higher (*p* < 0.0001) antibody titers to postfusion F BV2128 (GMT = 164,082) compared to titers in animals immunized with DS or DS-Cav1 (GMT = 3424 and 2177, respectively) (Figure 4B). Interestingly, animals immunized with prefusogenic F and DS, Cav1, or DS-Cav1 had equivalent antibody titers to prefusion F (BV2129) and DS-Cav1 (BV2280) (Figure 4C,D). These results demonstrate that prefusogenic F elicited high titer antibodies to prefusion and postfusion F structures, while cotton rats immunized with DS and DS-Cav1 produced antibodies primarily targeting prefusion structures.

### 3.5. Prefusion and Postfusion F Antibody Specificity

A competitive ELISA was used to determine the specificity of antibodies to prefusion and posfusion F conformers. Competition with prefusion F (BV2129) resulted in a 288-to 468-fold reduction in prefusion-specific antibodies in immune serum from animals immunized with DS or DS-Cav1, while competition with postfusion F (BV2128) resulted in only a < 3-fold reduction in antibody levels, consistent with the majority of antibodies being directed to the prefusion F structure (Figure 5A). In contrast, animals immunized with prefusogenic F or Cav1 produced antibodies that competed with both prefusion and postfusion F structures. Competition with prefusion resulted in a 34-fold reduction in prefusion-specific F antibodies, while competition with postfusion F resulted in a 13-fold reduction in antibodies, which indicates prefusogenic F and Cav1 elicited a mix of antibodies that bound both pre- and postfusion F structures (Figure 5A).

We next determined the conformation specificity of RSV/A neutralizing antibodies elicited by immunization with prefusogenic F compared to immunization with the prefusion F variants. Competition with prefusion F (BV2129) resulted in complete adsorption of neutralizing antibodies in the serum of cotton rats immunized with prefusogenic F, DS, and DS-Cav1 and near complete adsorption of neutralizing antibodies in serum of Cav1 immunized animals (Figure 5B). In contrast, adsorption with postfusion F (BV2128) resulted in partial or no loss of neutralizing antibodies. These results indicate that immunization with prefusogenic F, DS, Cav1, and DS-Cav1 elicited neutralizing antibodies that primarily recognize the prefusion F structure (Figure 5B).

### 3.6. RSV F Antigenic Site-Specific Antibodies

We next determined the specificity of the humoral response to key RSV F neutralizing epitopes with a panel of neutralizing mAbs using a competitive ELISA. Animals immunized with prefusion F DS, Cav1 or DS-Cav1 had significantly higher levels (*p* ≤ 0.001) of antibodies competitive with D25 (GMT = 40.8–76.2 µg mL^−1^) compared to levels in animals immunized with prefusogenic F (GMT = 9.0 µg mL^−1^) (Figure 6A). Importantly, animals immunized with prefusogenic F or Cav1 had higher levels (*p* < 0.0001) of antibodies competitive with palivizumab (site II) and R1.42 (site IV). In contrast, animals immunized with prefusion DS or DS-Cav1 had significantly lower (*p* ≤ 0.0001) levels of antibodies that were competitive with palivizumab (site II), R1.42 (site IV) or p27 (Figure 6B–D). These results are consistent with the mouse immunogenicity result and support the hypothesis that the flexible native prefusogenic F elicits antibodies that recognize neutralizing epitopes on prefusion and postfusion F conformations, while prefusion variants DS and DS-Cav1 produced antibodies primarily targeting prefusion site Ø but little or no antibodies to other neutralizing sites (II and IV).

### 3.7. Neutralizing Antibodies and Protection against RSV/A Challenge in Cotton Rats

RSV A and B subtype cross neutralizing antibody titers were determined in cotton rats immunized with prefusogenic F and prefusion F variants. RSV/A neutralizing antibody titers were significantly higher (*p* ≤ 0.04) in animals immunized with prefusogenic F or Cav1 (GMT = 2346 and 1594 IU mL^−1^, respectively) compared to titers in animals immunized with DS or DS-Cav1 (GMT = 532 and 282 IU mL^−1^, respectively) (Figure 7A). Significantly, animals immunized with prefusogenic F and Cav1 had 13- to 20-fold higher levels of neutralizing antibodies that cross-neutralized RSV/B (18537) subtype (GMT = 1559 and 1264 IU mL^−1^, respectively) while animals immunized with DS or DS-Cav1 had significantly lower (*p* ≤ 0.001) levels of RSV/B cross neutralizing antibody titers (GMT = 116 IU mL^−1^ and GMT = 79 IU mL^−1^, respectively) (Figure 7B).

To assess the protective effects of immunization with various RSV F protein constructs, three weeks after receiving two immunizations, cotton rats were challenged intranasally with 10^5^ pfu RSV/A Long. Virus load was significantly lower (*p* ≤ 0.005) in nose homogenates of all vaccinated animals (GMT = 1077–14,024 pfu g^−1^) compared to virus load in non-immunized animals (GMT = 558,308 pfu g^−1^). Furthermore, animals immunized with prefusogenic F had significantly lower (*p* < 0.01) virus load in nasal homogenates compared to animals immunized with the DS-Cav1 (GMT = 2587 pfu g^−1^ vs. GMT = 14,024 pfu g^−1^, respectively). Nasal virus loads were not significantly different among animals immunized with prefusogenic F, DS or Cav1 (GMT = 1077–9290 pfu g^−1^) (Figure 7C). Virus loads in lung homogenates were below the limit of detection (LOD < 100 pfu g^−1^) in lungs of animals immunized with RSV F proteins, and therefore were significantly lower than the virus load in placebo treated animals (GMT = 449,018 pfu g^−1^) (Figure 7D). Animals immunized with RSV F proteins had significantly reduced upper and lower respiratory tract virus load compared to virus load in non-immunized cotton rats.

## 4. Discussion

Prefusion stabilized F DS-Cav1 has been proposed to be a potentially superior vaccine candidate since high affinity neutralizing antibodies bind to conformation-dependent sites Ø and V [22,25,35]. Recent clinical studies using a site Ø mAb have indicated that prefusion-specific immunity can in fact protect against both RSV A and B subtypes [37], but as a strategy, the efficacy of a monovalent prefusion mAb as prophylaxis has also been shown clinically to be highly susceptible to failure due to the emergence of new resistance mutant strains [38]. Together this suggests that multiple F-specific neutralizing epitopes may be best employed to achieve protection. Importantly, in a recent global randomized placebo-controlled trial using the prefusogenic F vaccine (described herein) in the context of maternal immunization, a trial which did not meet the primary endpoints, there was clear evidence that the vaccine reduced severe RSV related disease and pneumonia of any cause [39]. Moreover, in the setting where trial conduct factors were favorable for data collection, the efficacy against severe RSV related hypoxia (74%, CI 50, 60) or hospitalization (59%, CI 37, 72) [39] Appendix A and a similar reduction was seen in high-risk infants given mAbs palivizumab or motavizumab, mAbs which target antigenic site II alone [3,4,40]. The overall trial results are consistent with the observation that higher maternally acquired RSV antibodies are associated with protection of infants against community-detected cases of RSV LRTI and pneumonia [41], and protection of infant baboons born to mothers immunized with prefusogenic F compared to infants born to non-immunized mothers [32]. We have previously shown that women of childbearing age immunized with prefusogenic F nanoparticles produce antibodies that bind multi-antigenic sites including sites II, IV, V and Ø [33,34,42,43] and we posit here that such an immune response appears to drive the protective efficacy seen in the Phase 3 trial.

Here, we compared the humoral immune response elicited by native prefusogenic F to prefusion F variants. Using a flexible prefusogenic F backbone, prefusion stable variants DS, Cav1 and DS-Cav1 were generated. Mice and cotton rats immunized with DS or DS-Cav1 had significant levels of antibodies targeting prefusion-specific sites Ø and V (Figure 3C, Figure 6A and Appendix A) but had little or no antibodies to key neutralizing antigenic sites II or IV (Figure 3D,E and Figure 6B,C). Immunization with prefusogenic F also elicited elevated levels of antibodies competitive with D25, although at lower levels than those elicited by immunization with prefusion variants (Figure 6A). In contrast, immunization with the flexible prefusogenic F elicited antibodies that recognized both prefusion and postfusion F structures (Figure 5A). Importantly, immunization with prefusogenic F produced antibodies that broadly cross-neutralized RSV A and B subtypes, while immunization with prefusion DS or DS-Cav1 elicited RSV/A neutralizing antibodies but few RSV/B cross-neutralizing antibodies (Figure 7A,B). Collectively, these results show that prefusogenic F was significantly more immunogenic and the flexible structure allowed B-cell access to important neutralizing epitopes that were not available on more ridgid F structures.

The concept of type 1 trimer breathing, dissociation, and monomerization on the lipid bilayer has been described for the human immunodeficiency virus (HIV) envelope (Env) glycoproteins gp120 and gp41. Lee et al. [44,45,46] have shown that antibody binding to flexible Env gp promotes the dissociation of the Env gp120/gp40, which promotes trimer instability and may be an unrecognized mechanism of protection. These observations support the concept that Env immunogens that closely mimic the flexible Env gp structure may be critical considerations in vaccine design, allowing B-cells to access Env epitopes that promote trimer instability and decay. This same mechanism may also be important when designing RSV F trimeric vaccines. Gillman et al. [19] recently reported that CR9501 mAb binding to RSV F at a specific angle promotes trimer disassembly within the lipid bi-layer. These observations suggest that vaccine strategies targeting type 1 trimers should consider flexible trimeric structures and the plasticity of humoral immunity. Such vaccine strategies may be superior to utilizing rigid structures.

## 5. Conclusions

In this report, we hypothesized that trimeric vaccine strategies based on breathable F structures allow B-cells to access epitopes on intermediate conformations, which may collectively contribute to broad protective immunity, while less flexible F structures restrict B-cell access to important protective epitopes. We compared the humoral immune responses to prefusogenic F to prefusion stabilized variants. Native prefusogenic F was significantly more immunogenic and elicited antibodies to conformation-dependent and conformation-independent epitopes that broadly cross-neutralized RSV A and B subtypes, while prefusion variants DS and DS-Cav1 were less immunogenic and elicited antibodies primarily targeting the prefusion F structure. These results have implications for vaccine strategies targeting other type 1 integral membrane proteins.

## Figures and Tables

**Figure 1 vaccines-08-00607-f001:**
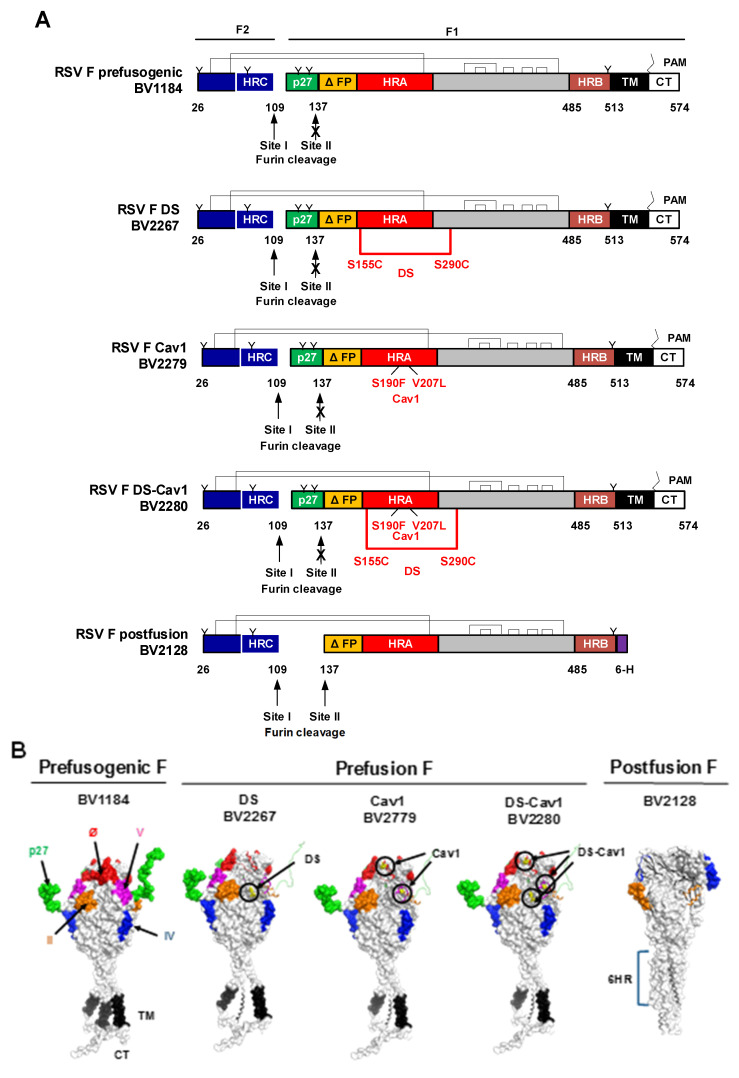
Respiratory syncytial virus fusion protein (RSV F) constructs used in this study. (**A**) Linear diagram of the RSV F glycoproteins constructed from the wild-type RSV/A2 fusion glycoprotein gene sequence (GenBank accession number MN125707). RSV F has two furin cleavage sites (arrows) that flank the intervening 27 amino acids (p27 fragment, green). To produce the prefusogenic RSV F (BV1184), furin cleavage site II was mutated (KKRKRR → KKQKQQ) to be protease resistant and is retained on the N-terminus of the F1 subunit. The first 10 amino acids of the adjacent fusion peptide were deleted (ΔFP, yellow). The native native transmembrane (TM, black) and cytoplasmic tail (CT, white) were retained on the C-terminus of F1. The resulting F2 and F1 ectodomain is covalently coupled by two disulfide bonds (brackets). *N*-linked glycosylation sites are indicated by “Y”; palmitoleic acid (PAM) is indicated by “ϟ”; heptad repeats A, B, and C are indicated by HRA (red), HRB (brown) and HRC (blue). The prefusogenic F structure (BV1184) backbone was used to produce three prefusion F variants. RSV F DS (disulfide bond) variant (designated BV2267) was generated by introducing a stabilizing disulfide bond at positions S155C and S290C within the F1 subunit. RSV F Cav1 (BV2279) construct was generated by introducing two hydrophobic amino acids at positions S190F and V207L. The RSV F DS-Cav1 construct (BV2280) was generated with S155C/S290C disulfide bond and the S190F/V207L substitutions. Postfusion F (BV2128) was generated with intact furin cleavage sites I and II and truncation of the fusion peptide (ΔFP, yellow). The TM and CT domains were deleted and replaced with a 6-histidine tag (6-H, purple). (**B**) Surface structures of the RSV F prefusogenic and prefusion trimers indicating key neutralizing antigenic sites antigenic sites Ø (red), V (pink), II (orange), and IV (blue), and the p27 fragment (green). The native transmembrane (TM, black) and cytoplasmic tail (CT, grey) are indicated. The RSV F postfusion 6-heptarepeat bundle (6HR) is indicated.

**Figure 2 vaccines-08-00607-f002:**
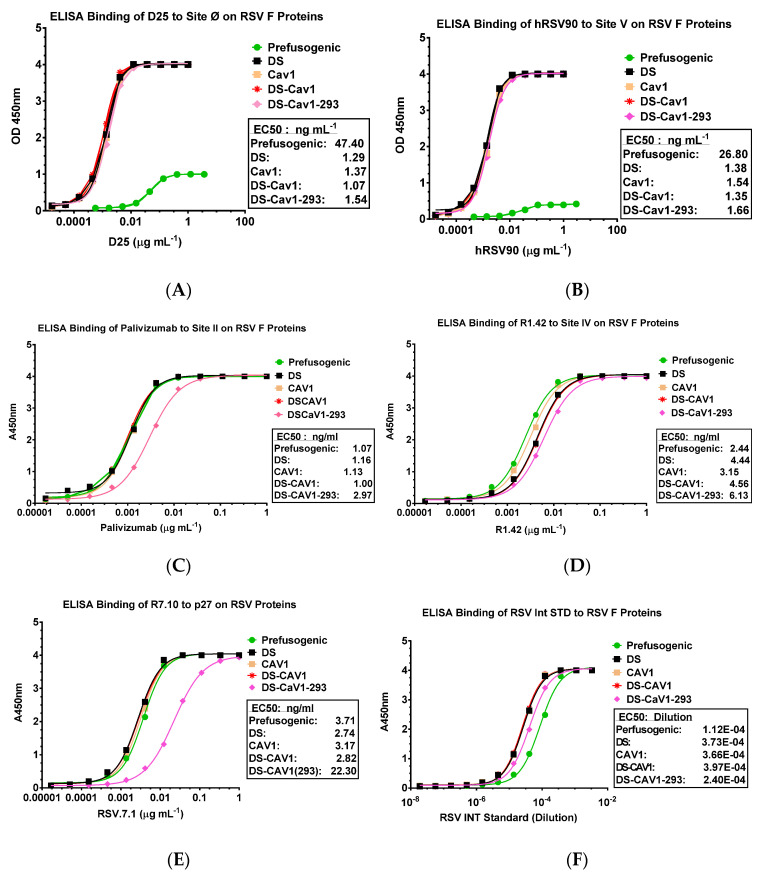
Antigenic site-specific monoclonal antibody binding to prefusogenic F and prefusion F variants. Enzyme-linked immunosorbent assay (ELISA) binding curves monoclonal antibodies (**A**) D25 (site Ø), (**B**) hRSV90 (site V), (**C**) palivizumab (site II), (**D**) R1.42 (site IV), (**E**) R7.10 (p27), and (**F**) human RSV international standard (RSV Int Std).

**Figure 3 vaccines-08-00607-f003:**
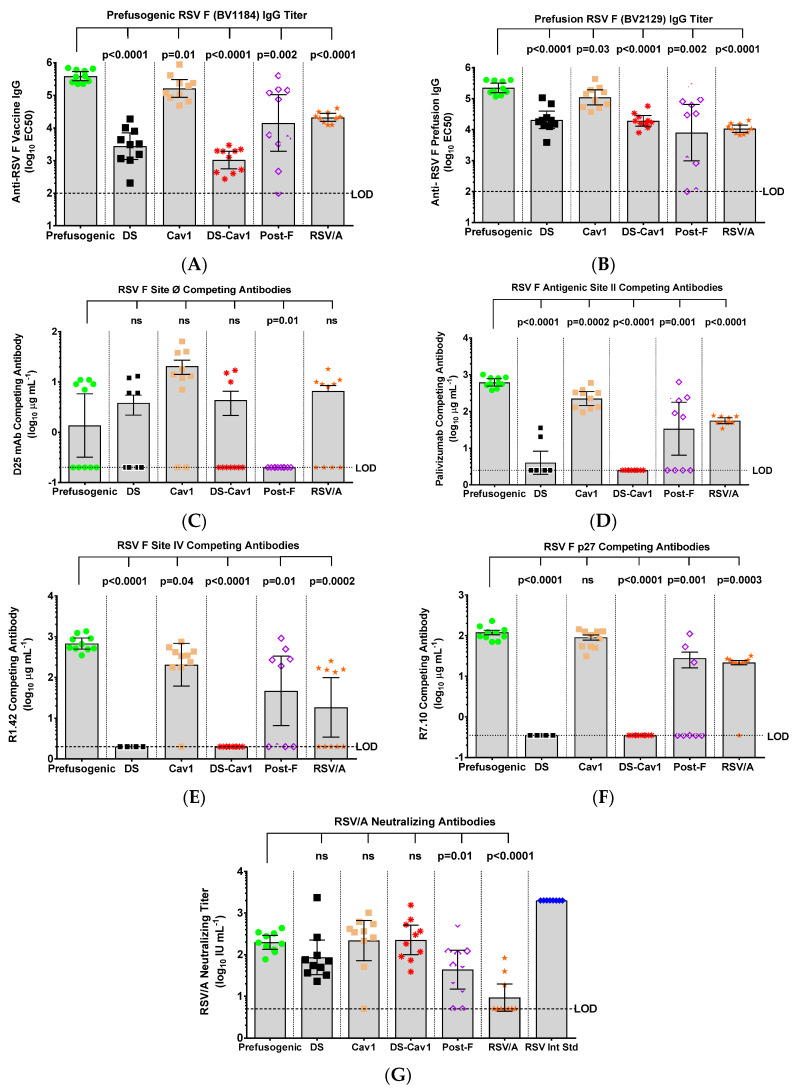
Immunogenicity of prefusogenic F and F variants in mice. Groups of mice were immunized by intramuscular injection with prefusogenic F vaccine (BV1184), DS (BV2267), Cav1 (BV2279), DS-Cav1 (BV2280) or postfusion F (BV2128) with 30 µg aluminum phosphate on days 0 and 21. A separate group (RSV/A) was challenged intranasal with 10^6^ plaque forming units (pfu) RSV/A Long strain. Serum was collected for analysis from all animals on study day 35 (14-days after the second immunization). (**A**) Anti-prefusogenic F vaccine IgG (BV1184) and (**B**) anti-prefusion F IgG (BV2129) determined by ELISA. A competitive antibody ELISA was used to determine serum levels of antibodies that competed the binding of site-specific monoclonal antibodies (mAbs) (**C**) D25 (site Ø), (**D**) palivizumab (site II), (**E**) R1.42 (site IV), and (**F**) R7.10 (p27). (**G**) RSV neutralizing antibody titers were determined by inhibition of HEp-2 cell infection with RSV/A Long. Symbols indicate the antibody titer for individual animals in each immunization group. Bars indicate the group (N = 10 per group) geometric mean titer (GMT) and the error bars indicate the 95% confidence interval (±95% CI). Significance between the RSV F prefusogenic vaccine (BV1184) group and paired RSV F protein immunized groups is indicated. Limit of detection (LOD). Not significant (ns).

**Figure 4 vaccines-08-00607-f004:**
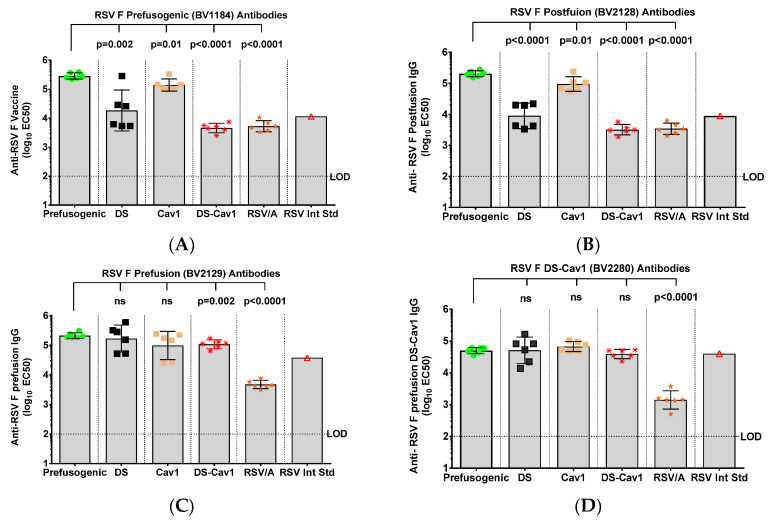
Immunogenicity of prefusogenic F and F variants in cotton rats. Cotton rats were immunized by intramuscular injection with 1.0 µg of prefusogenic F vaccine (BV1184), DS (BV2267), Cav1 (BV2279), DS-Cav1 (BV2280) or postfusion F (BV2128) with 30 µg aluminum phosphate on days 0 and 21. A comparative control group was intranasal challenged with 10^6^ pfu RSV/A Long. Serum was collected for analysis 21-days after the second immunization (day 42). (**A**) Anti-prefusogenic F vaccine IgG (BV1184). (**B**) Anti-postfusion F IgG (BV2128). (**C**) Anti-prefusion F IgG (BV2129). (**D**) Anti-DS F IgG (BV2280). Not significant (ns). Symbols indicate antibody titers for individual animals in each immunization group. Bars indicated the geometric mean titer (GMT) and the error bars indicate the ± 95% CI. Statistical significance between the prefusogenic F (BV1184) compared to paired groups. Limit of detection (LOD). Not significant (ns).

**Figure 5 vaccines-08-00607-f005:**
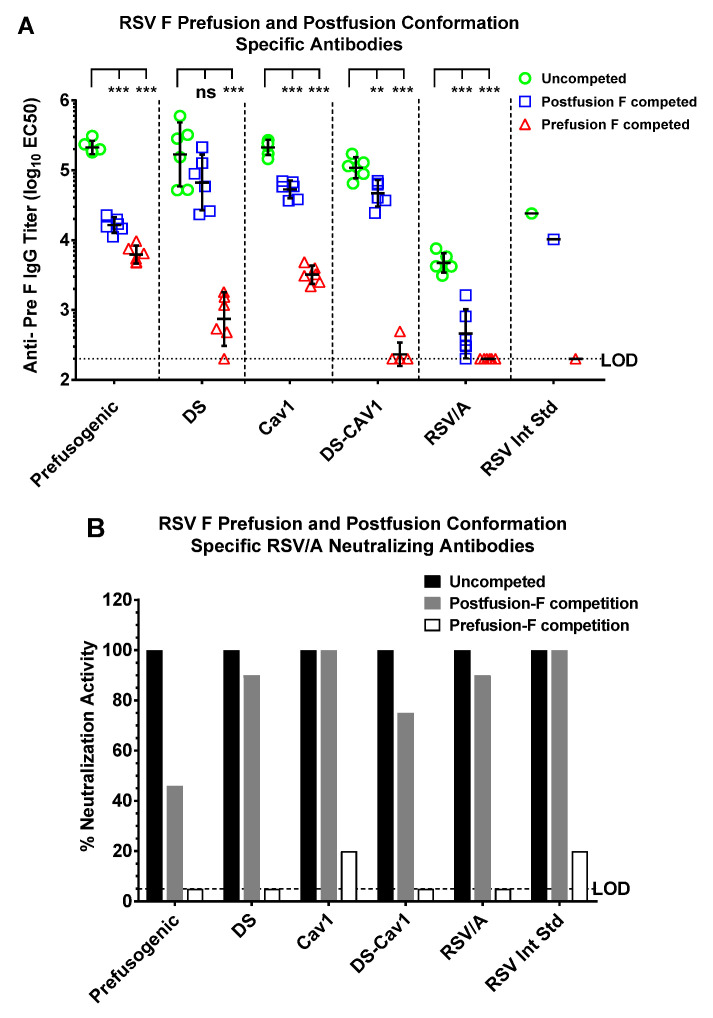
Prefusion and postfusion F specificity of antibodies elicited by prefusogenic F and prefusion F variants. Cotton rats were immunized on days 0 and 21, and serum analyzed 21 days after the second immunization (day 42) as described in Figure 4. (**A**) A competitive ELISA was used to determine the level of antibodies targeting prefusion F (BV2129) and postfusion F (BV2128) in serum of immunized cotton rats. Individual animal titer is indicated by the colored symbol for each immunization group. The geometric mean titer (GMT) is indicated by the horizontal bar and the error bars indicate the 95% CI. (**B**) Conformation-specific neutralizing antibodies in serum competed with prefusion (BV2129) or postfusion F (BV2128). The group (N = 6 per group) GMT is indicated by the solid bars. Lower limit of detection (LOD). ** *p* = 0.003, *** *p* < 0.0001, not significant (ns).

**Figure 6 vaccines-08-00607-f006:**
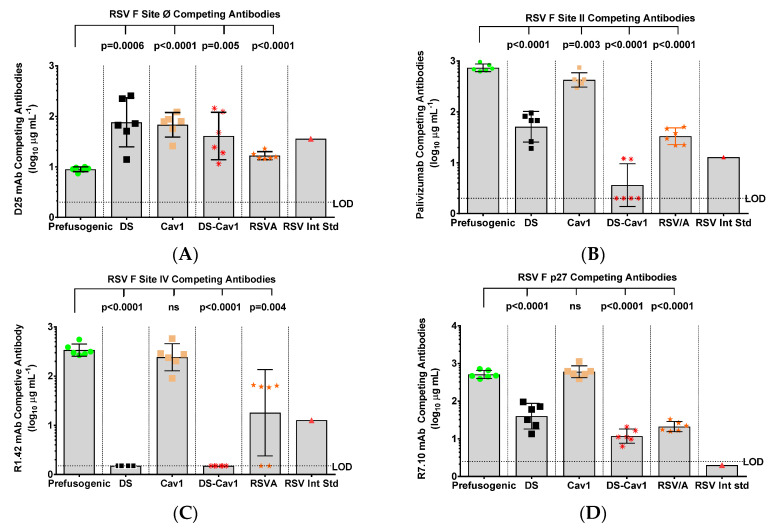
Specificity of antibodies elicited by prefusogenic F compared to prefusion F variants in cotton rats. Female cotton rats were immunized with 1.0 µg RSV F proteins with 30 µg aluminum phosphate and a control group was challenged intranasally with RSV/A Long. Cotton rats were immunized and serum analyzed 21 days after the second immunization, as described in Figure 4. The specificity of the humoral response was determined by a competitive antibody binding ELISA using site-specific mAbs: (**A**) D25 (site Ø), (**B**) palivizumab (site II), (**C**) R1.42 (site IV), and (**D**) R7.10 (p27). Individual animal values are indicated by colored symbols for each immunization group. The bars indicate the group (N = 6 per group) geometric mean titer (GMT) and the error bars indicate the ± 95% CI. Statistical significance between the RSV prefusogenic F group (BV1184) and the paired groups is indicated. World Health Organization (WHO) RSV International Standard (RSV Int Std). Limit of detection (LOD). Not significant (ns).

**Figure 7 vaccines-08-00607-f007:**
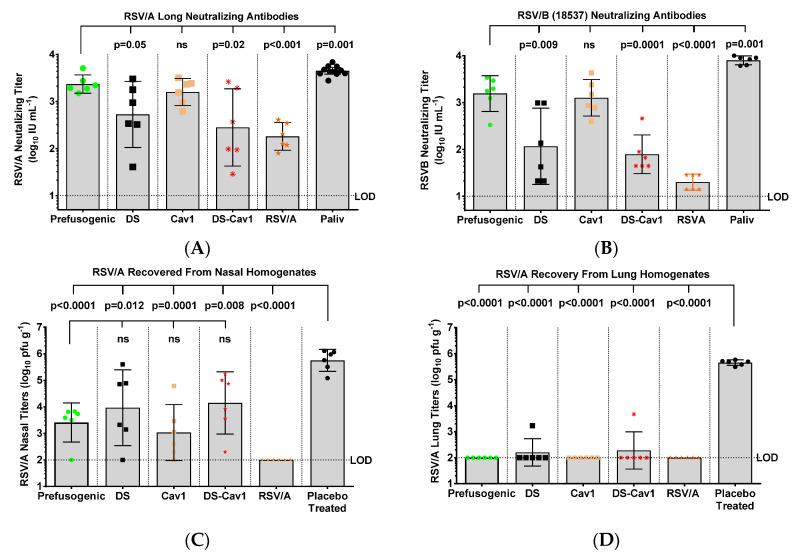
RSV A and B subtype neutralizing antibodies and protection against RSV challenge in cotton rats. (**A**,**B**) RSV A and B subtype cross-neutralizing antibody titers in animals immunized with prefusogenic F and prefusion F DS, Cav1 and DS-Cav1. (**C**,**D**) Nose and lung tissues were collected 4-days post-challenge (day 46) and infectious virus determined with a plaque assay. Individual animal antibody titers and virus load in nasal and lung homogenates are plotted. Symbols indicate individual animal values for each immunization group. Bars indicate the group geometric mean titer (GMT) and the error bars indicate the ±95% CI. Statistical significance between paired groups is indicated by brackets. Limit of detection (LOD). Not significant (ns). Palivizumab (Paliv).

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
