# Peer review of "Flexible RSV Prefusogenic Fusion Glycoprotein Exposes Multiple Neutralizing Epitopes that May Collectively Contribute to Protective Immunity"

_vaccines, 2020, doi:10.3390/vaccines8040607_

Round 1
Reviewer 1 Report
The manuscript entitled Flexible RSV prefusogenic fusion glycoprotein exposes multiple neutralizing epitopes that may collectively contribute to protective immunity by Patel et al. presents results showing that prefusogenic F is significantly more immunogenic then prefusion Ds and DS-Cav1 F variants. The authors also propose that flexible structure of prefusogenic F allows B-cell access to important neutralizing epitopes that are not available at more ridged F structures.
I think this is a nice study. Overall, the manuscript reads well and includes relevant citations and depth.
Minor comment:
Concerning the general view of the manuscript, even the study focuses on humoral immune response, the authors could provide experimental evidence or at least discuss if there are or might be any differences on proinflammatory mediators (such as IL-17, IFNγ regulated protein, MCP-1, RANTES, …) produced and released in association with RSV challenge in Balb/c mice or cotton rats immunized by various RSV F variants. Similarly, the analysis of the level of expression of transcripts encoding cytokine/chemokine genes in lung tissue isolated from mice or cotton rats immunized by various RSV F variants and exposed to RSV, might be highly informative.
Additionally, the role of toll-like receptors in receiving and transducing proinflammatory signals should be discussed within the manuscript, including the interplay of macrophages, dendritic cells, NK cells, and T lymphocytes in modulating the inflammatory response to RSV infection.
High-resolution Figure 1B showing surface structures of the RSV F prefusogenic and prefusion trimers should be provided.
Reviewer 2 Report
Patel et al have conducted a series of experiments to determine the immunogenicity of candidate RSV fusion proteins to determine which may provide optimal protection from RSV infection. With no licensed RSV vaccine available, studies such as these are absolutely necessary. The authors compare mutant forms of the fusion (F protein) in their studies. The different forms are depicted nicely in Figure 1 to aid in clarity for the reader. Essentially the authors compare a “breathable” form of the F protein, termed prefusogenic, to other similar constructs that have mutations which restrict the ability of the F protein to change its confirmation. The authors conclusion that the prefusogenic F protein elicits a broader antibody response is largely supported by the data but with some limitations that the authors take care to mention.
I think it will be important for the authors to emphasize that despite the broader repertoire of antibodies elicited, protection of animals experimentally infected with the virus following vaccination is unaltered when comparing breathable to static F protein constructs (Figure 7C and D) and that neutralizing antibodies are equivalent in immunized mice while differences are observed in cotton rats. These data would suggest that perhaps the advantages of using a prefusogenic F protein as a vaccine candidate may not yield better results than similar, more rigid, forms of the protein.
My only concerns with this paper are largely editorial. A detailed proof reading of the manuscript would have been useful as in many instances, words are missing from sentences which confuse the reader. I will give a few examples, but strongly ask the authors to take the time to re-read the manuscript to fix additional mistakes I missed.
Line 48 newborn should be newborns
Line 307 “Prefusogenic F nanoparticles (BV1184) also by bound by mAbs D25” needs to be corrected
Line 333. Last sentence should be “Mice immunized with…” instead of “Immunized with…”
Line 431 “Animals immunized with prefusion F DS, Cav1” I believe the F is a typo
Line 464 Immunizations not immunization. That or delete the “2” in the sentence
Line 465 virus load was significantly lower “in” nose homongeates
There are also a few phrases that are confusing and should be edited.
Line 48-49. This sentence seems could use additional context, correlates of protective immunity in what?
Line 527 prefusion variants could use some additional adjectives, such as static or constrained to further signify what you are discussing
And I would advise the authors to perhaps include more descriptive labels in the figures. For instance, what is the difference between Figures 3a and 3b. I could not figure this out for a long time and had to re-read the materials and methods section twice and I still am not 100% sure I have it correct.
